# HyDance: A Novel Hybrid Dance Generation Network with temporal and frequency features

## Abstract

We propose HyDance, a transformer based diffusion network utilizing both the temporal and frequency domain representations of dance sequences for music-driven dance generation. Existing dance generation methods primarily use temporal domain representations of dances, which results in the network losing the frequency domain characteristics of the dance sequences. This manifests in overly smooth generated dance sequences, resulting in dance movements that lack dynamism. From an aesthetic perspective, such overly smooth movements are perceived as lacking expressiveness and the sense of power. To address this issue, we propose HyDance, which incorporates independent temporal feature encoders and frequency domain feature encoders. The model employs a shared-weight hybrid feature encoder, enabling the complementary extraction of motion information from both domains. By introducing compact frequency domain representation into the dance generation framework, our method mitigates the oversmoothing problem in generated dance sequences and achieves improved spatial and temporal alignment in the generation results. Experiments show that our method generates more expressive dance movements than existing methods and achieves better alignment with the music beats.

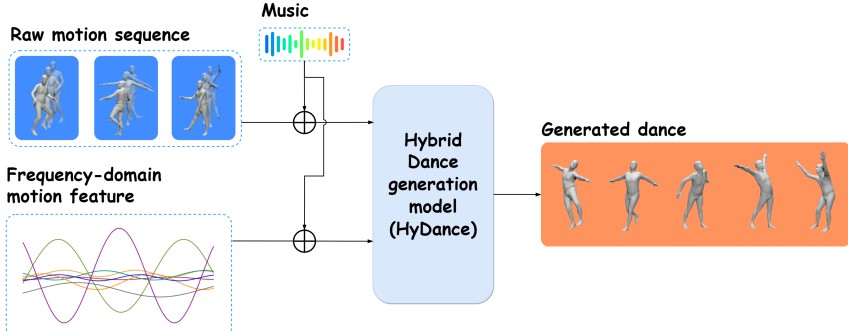

Figure 1: HyDance generates physically plausible and smoothly flowing dances conditioned on music. During the training stage, our model utilizes dance sequences represented in two different forms, which allows for better capture of motion details and results in more realistic dance generation compared to previous methods.

## 1 Introduction

In the field of human motion synthesis, dance synthesis represents a specialized research direction worthy of exploration. As an art form, dance contains rich semantic information derived from human aesthetics, and a method capable of effectively generating high-quality dance sequences that conform to specified conditions possesses a wide range of applications, including content creation areas such as video production and game cutscenes.

Furthermore, in terms of motion characteristics, dances feature a diverse range of movements. Compared to common human motions such as walking or running, the differences between various dance movements are more pronounced, and transitions between motions are more frequent. This includes a substantial number of transitions between high-frequency and low-frequency dance movements, which presents a greater challenge to the generation capabilities of the model.

In this paper, we aims to generate detailed 3D dance sequences that conform to human aesthetic standards of dance movements, given a piece of music. In recent years, with advancements in generative model technologies, researchers have made significant progress in the field of dance sequence synthesis. However, previous work primarily relies on the temporal joint rotation representation during model training, which limits the model's performance due to the sparsity of motion representation. As a result, the generated dance sequences often exhibit overly smooth characteristics, lacking the fine dynamics present in real dance performances. In the field of motion representation learning, in addition to directly using joint rotation representations, structured representations learn to extract essential features from the motions, thereby creating a more compact feature space. Such representations generally help to improve the dynamic performance of the motion sequences generated by the model. Recent work named PAE (Starke et al., 2022) proposes using an autoencoder structure, introducing frequency domain transformations as inductive bias to capture the periodic features of human motion from unstructured motion sequences. This approach has achieved good results in synthesis tasks with relatively simple motion categories, producing flexible and lively motions with smooth transitions between high and low-frequency actions. However, for dances, encoding only the periodic features of the dance sequences is not sufficient to fully restore the intricate details of the dance performance.

In terms of model design, while the capabilities of transformer models have been validated across many fields, recent research(Piao et al., 2024; Park & Kim, 2022; Guo et al., 2023; Tian et al., 2023) has highlighted certain learning biases inherent in transformer models. This issue specifically refers to the fact that the self-attention mechanism tends to capture low-frequency features in the data and can easily overlook high-frequency features. Researchers, through in-depth studies of the mechanisms of transformers, have found that self-attention layers often exhibit low-pass filtering characteristics, tending to capture tokens that appear consistently in the context. For dance sequence data, high-frequency motion features typically appear at the transitions between dance movements or in the detailed movements of limbs within a segment of dance. Since these movements are not choreographically designed and the transitions between dance movements can vary widely, they do not frequently occur in dance sequences. This means that high-frequency features of dance sequences usually do not affect the overall direction or general trajectory of the dance movements but do impact the expressiveness and the sense of power in the dance movements. The relatively low energy of high-frequency features in the dance generation results demonstrates the oversmoothing of dance in the frequency domain.

To address this issue, we propose HyDance, a novel dance generation framework that combines frequency domain representation with temporal joint rotation representation of dance sequences to generate high-quality dance sequences. We follow PAE to obtain the motion manifolds of dances and then use it to build the frequency domain representations of dance sequences. Because of the more compact feature space in learned motion manifolds of dances, integrating this frequency domain representation helps the model reduce the probability of sampling meaningless samples. To compensate for the limitations of frequency domain representation in encoding non-periodic dance movement features, we design the model to accept both frequency domain motion representation and temporal joint rotation motion representation, thereby allowing the model to acquire complementary motion information from both representations. The generated dance sequences achieve richer transitions and are more expressive compared to existing methods.

To summarize, our contributions include the following:

1) We introduce the frequency domain representation of dance sequences into the dance generation framework, proposing HyDance, a dance generation framework that combines frequency domain and temporal representations for high-quality dance generation.

2) A Dual-Domain Hybrid Encoder that helps the dance generation framework effectively utilize the additional information to optimize the dance generation results.

3) Experiments demonstrate that our method generates dances that better align with human aesthetic standards, featuring more expressive movements and more natural transitions.

## 2 RELATED WORK

### 2.1 HUMAN MOTION SYNTHESIS

Human motion synthesis is a topic of significant interest among researchers in computer vision. Earlier motion synthesis methods typically used GAN models. (Habibie et al., 2022) employs motion matching techniques to acquire reference motion sequences and refines these motion sequences through a GAN model to synthesize body pose sequences that conform to speech. (Hernandez et al., 2019) uses past motion sequences and leverages a GAN model to predict subsequent motion sequences, achieving consistent body poses and root translations in the synthesized motion sequences. Additionally, VAEs have been widely used in many works for motion synthesis. (Guo et al., 2022) proposes using motion snippet codes as motion representations within the model and employs a VAE model to achieve text-to-motion generation.

In recent years, as diffusion models have demonstrated powerful generation capabilities, there has been significant progress in numerous works that employ these models. (Tevet et al., 2023) achieves Text-Conditioned Motion Synthesis with a Diffusion framework and obtains high-quality motion generation results. (Dabral et al., 2023) uses a diffusion-based motion generation framework and can leverage textual or musical conditioning information to control the generation process, offering multimodal generation capabilities. (Chen et al., 2023) first maps motions to low-dimensional latent codes using a VAE and then uses these latent codes as input to a diffusion model, thereby achieving motion synthesis with lower computational requirements. (Zhang et al., 2024b) separately constructs global trajectory and local pose modules, leveraging the generation capabilities of diffusion models to achieve 3D motion reconstruction from monocular video frames.

### 2.2 MUSIC DRIVEN DANCE GENERATION

Synthesizing dance sequences that match the style and rhythm of given music and have high motion quality is a topic of significant interest among researchers. Various methods, including sequence-based models, GANs, VAEs, and diffusion models, have been proposed to tackle this task. Sequence-based dance generation models usually employ transformer-based models to generate dance sequences in an autoregressive manner. (Li et al., 2021) utilizes a transformer-based model, taking initial short motion sequences and music features as input, and generates dance sequences by predicting the next motion frame-by-frame in an autoregressive manner. (Sun et al., 2022) introduces a bank of latent codes to constrain generated dance sequences to remain close to ground truth dance sequences, thereby reducing error accumulation due to the autoregressive generation approach in sequence-based models and alleviating the motion freezing problem when generating long dance sequences. (Kim et al., 2022) uses a GAN architecture with a transformer encoder as the discriminator and a transformer decoder as the generator, enabling style-controlled dance generation through the use of style codes. (Siyao et al., 2022) first uses a VQ-VAE to map motions to a codebook, and then builds a GPT-based model to arrange motion codes according to control information and convert them into dance sequences. (Tseng et al., 2023) introduces a Diffusion model for dance generation, and (Li et al., 2024) achieves a balance between global choreographic patterns and local motion quality through a coarse-to-fine two-step diffusion framework. (Zhang et al., 2024a) divides the complete noisy motion sequence into short segments and uses a cross-attention mechanism to fuse information from preceding and following short segments during the denoising process, generating more coherent dance sequences. (Bhattacharya et al., 2024) achieved the generation of dance sequences that closely align with musical beats by separately generating short sequences of beat poses corresponding to the music beats and repetition poses between these beat pose sequences.

## 3 METHOD

### 3.1 PRELIMINARIES

**Diffusion model.** Diffusion models(Ho et al., 2020), a class of generative models, have gained significant attention due to their ability to generate high-quality samples. We follow DDPM (Ho et al., 2020) and built a transformer-based model to generate dance sequences. Diffusion models consist of two primary process, the forward diffusion process and the reverse denoising process. The forward diffusion process is defined as a Markov noising process operate by gradually adding noise $\{z_t\}_{t=0}^{T}$to the initial data $d_0$ according to a predefined schedule, which can be simplified into one step as:

$$q(z_t|d_0) = \mathcal{N}(\sqrt{\overline{\alpha}_t}d_0, (1 - \overline{\alpha}_t)I) \tag{1}$$

where $\overline{\alpha}_t \in (0, 1)$ is accumulated noise decay factors. As $t$ gradually increases, $\overline{\alpha}_t$ converges to zero, allowing us to approximate the result as $z_T \sim \mathcal{N}(0, 1)$. The reverse denoising process typically employs a neural network, which in our case is a Transformer-based model, to progressively remove the noise and thereby recover the data $\hat{d}_0$. By incorporating classifier-free guidance(Ho & Salimans, 2022), we can generate dance sequences that conform to the musical conditions.

**frequency domain feature extractor.** We follow the basic network structure of PAE(Starke et al., 2022) to construct our frequency domain feature extractor. PAE uses stacked 1D convolution layers to transform the original joint rotation representations of dance sequences into latent variables $w$ that have $m$ channels. It then calculates the latent parameterization $f, a, b, s$ for each latent channel using a differentiable FFT layer. During the decoding process, these latent parameterization along with the motion sequence window length $\mathcal{T}$ are used to reconstruct the latent variables,

$$\hat{w} = a \cdot \sin\left(2\pi \cdot (f \cdot \mathcal{T} - s)\right) + b \tag{2}$$

Then, we use stacked 1D convolutional layers to decode the latent variables back into motion sequences. After the training of frequency domain feature extractor is completed, we use a sliding window approach to apply the extractor to the full dance sequence, obtaining the latent parameterization $f, a, b, s$ of motion clips within each window. The latent parameterization is then used as the encoding result for the center motion frame of each window. Through this approach, we encode the original motion sequence into a structured frequency domain representation.

**Representation.** For a music sequence of $L$ frames, we follow (Li et al., 2021) and extract music features $m \in \mathbb{R}^{L \times 35}$ using librosa (Mcfee et al., 2015). The extracted music features consist of 35 channels, including a 1-dim envelope, 20-dim MFCC, 12-dim chroma, 1-dim one-hot peaks, and 1-dim one-hot beats.

For dance sequences, we use both temporal and frequency domain motion representations. For the temporal domain representation, we follow (Tseng et al., 2023) and represent the motion sequence as $d \in \mathbb{R}^{L \times 151}$, which includes: (1) 4-dim contact label indicating whether the heels and toes of the left foot and right foot are in contact with the ground; (2) 3-dim root translation; (3) 144-dim joint rotation, represented in a 6-DOF (Zhou et al., 2019) format. For the frequency domain representation, we represent the dance sequences as $d^f \in \mathbb{R}^{L \times 32 \times 4}$, which includes 32 channels of latent variables. Each latent channel contains four parameters $f, a, b, s$ per motion frame, representing the latent parameterization of the motion sequence segment within the window centered on the current frame.

### 3.2 DUAL-DOMAIN DANCE GENERATION FRAMEWORK

Given a music feature sequence, our goal is to generate a dance sequence $d_g \in \mathbb{R}^{L \times 151}$. We construct a Transformer-based diffusion framework that includes separate temporal and frequency domain encoders, a shared-weight hybrid encoder, and a dance decoder. The hybrid encoder allows the model to acquire complementary motion information from different representations, thereby enabling our framework to achieve higher quality dance generation.

### 3.3 LOSS

**Loss for training frequency domain feature extractor.** We divide the training of our overall dance generation framework into two parts. The first step is the training of the frequency domain feature

extractor, and the second step is the training of the dance generation model. During the training of the frequency domain feature extractor, for a dance sequence $\boldsymbol{d}$ we obtain the reconstructed dance sequence $\hat{\boldsymbol{d}}$ through the encoding and decoding process as described in chapter 3.1, and calculate the reconstruction error as follows:

$$\mathcal{L}_{recon} = \mathbb{E}[||\boldsymbol{d} - \hat{\boldsymbol{d}}||_2^2] \tag{3}$$

we observed that using only a simple reconstruction loss would lead to a significant loss of non-periodic motion features in the reconstructed motion sequences. Therefore, we introduced a part of the geometric loss similar to (Tevet et al., 2023) to encourage the joint positions of reconstructed motion sequences to match the joint positions of the original motion sequences. $FK(\cdot)$ denotes the forward kinematic process that converts joint rotations into joint positions and $(i)$ denotes the index of motion frame.

$$\mathcal{L}_{joint} = \frac{1}{N} \sum_{i=1}^{N} ||FK(\boldsymbol{d}^{(i)}) - FK(\hat{\boldsymbol{d}}^{(i)})||_2^2 \tag{4}$$

**Loss for training dance generation model.** During the training of the dance generation model, our model learns to estimate $\hat{\boldsymbol{d}}$ to reverse the forward diffusion process. We denote the model parameters as $\theta$ and optimize these parameters using the loss function introduced by (Ho et al., 2020), and we denote it as $\mathcal{L}_{simple}$. We follow (Tseng et al., 2023) by not only using a simple reconstruction loss but also incorporating multiple auxiliary losses, including joint position loss $\mathcal{L}_{joint}$ as shown in Equation 4, velocity loss $\mathcal{L}_{vel}$, and foot velocity loss $\mathcal{L}_{contact}$.

$$\mathcal{L}_{simple} = \mathbb{E}_{\boldsymbol{d},t}[||\boldsymbol{d} - \hat{\boldsymbol{d}}_\theta(\boldsymbol{z}_t, t, \boldsymbol{m})||_2^2] \tag{5}$$

$$\mathcal{L}_{vel} = \frac{1}{N-1} \sum_{i=1}^{N-1} ||(\boldsymbol{d}^{(i+1)} - \boldsymbol{d}^{(i)}) - (\hat{\boldsymbol{d}}^{(i+1)} - \hat{\boldsymbol{d}}^{(i)})||_2^2 \tag{6}$$

$$\mathcal{L}_{contact} = \frac{1}{N-1} \sum_{i=1}^{N-1} ||(FK(\hat{\boldsymbol{d}}^{(i+1)}) - FK(\hat{\boldsymbol{d}}^{(i)})) \cdot \hat{\boldsymbol{b}}^{(i)}||_2^2 \tag{7}$$

where $\hat{\boldsymbol{b}}^i$ represents the binary foot contact labels predicted by the model. The foot velocity loss $\mathcal{L}_{contact}$ enhances the physical plausibility of the generated dance sequences by penalizing the foot velocities in frames where the feet should be stationary. These losses encourage the generated results to be more physically plausible from different perspectives.

In addition, we introduce an extra motion decoding loss as shown in Equation 8. In the network, the frequency domain representation of the motion sequence $d^f$ passes through the motion decoder to obtain the reconstructed temporal representation of the dance sequence $\hat{d}^f$, allowing the hybrid encoder to effectively integrate frequency domain and temporal motion representations, and enabling the motion decoder to utilize the unified motion representation to decode the dance sequence.

$$\mathcal{L}_{f2m} = \mathbb{E}_{\boldsymbol{d},t}[||\boldsymbol{d} - \hat{\boldsymbol{d}}^f(\boldsymbol{z}_t, t, m)||_2^2] \tag{8}$$

In summary, our overall training loss is expressed as a weighted sum of the aforementioned losses, as shown below:

$$\mathcal{L} = \lambda_{simple}\mathcal{L}_{simple} + \lambda_{joint}\mathcal{L}_{joint} + \lambda_{vel}\mathcal{L}_{vel} + \lambda_{contact}\mathcal{L}_{contact} + \lambda_{f2m}\mathcal{L}_{f2m} \tag{9}$$

## 3.4 MODEL

Our dance generation framework is illustrated in Fig. 2. We incorporated the pre-trained frequency domain feature extractor into our framework. During training, we first use the frequency domain feature extractor to compute the frequency domain representation of the motion sequence. Then, we obtain the noisy temporal and frequency domain representations $\boldsymbol{d}_t, \boldsymbol{d}_t^f$ of the motion sequence through a forward diffusion process. Along with the music condition $\boldsymbol{c}$ and time step $t$, we input $\boldsymbol{d}_t, \boldsymbol{d}_t^f$ into the model together. The motion sequences in both representations are processed separately through independent temporal and frequency domain encoders with self-attention layers, followed by a shared-weight hybrid encoder. We aim to map both representations of dance sequences into a common feature space, enabling the motion decoder to incorporate motion information from

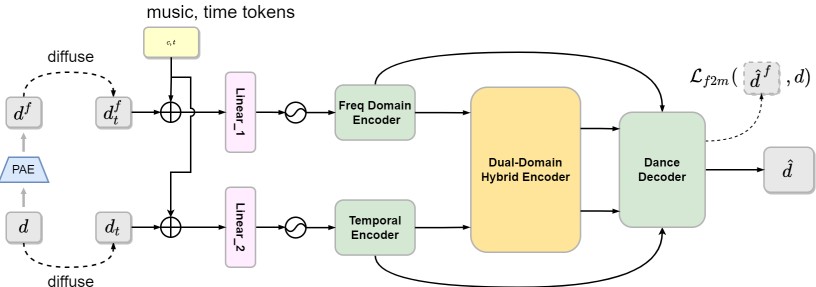

Figure 2: An overview of our HyDance framework. HyDance learns to generate dance sequences conditioned on music by processing dance sequences in two different representations.

both the frequency domain representation and the temporal joint rotation representation to generate dance sequences. To achieve this, we designed a hybrid encoder, as shown in Fig 3. Both representations of dance sequences are mapped to a common feature space through the hybrid encoding process. Finally, the noisy motion sequences in two different representations, are fed into the dance decoder together with the unified motion feature sequences obtained from the hybrid encoder. The motion decoder includes cross-attention layers and FiLM(Perez et al., 2018) layers to obtain the generated dance sequence $\hat{d}$.

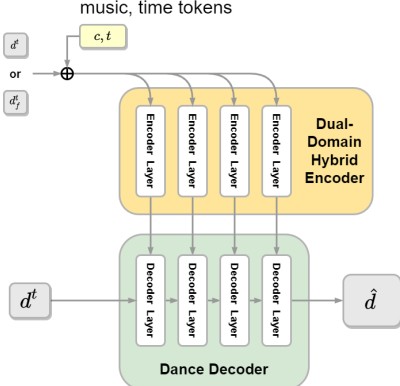

Figure 3: An illustration of the Dual-Domain Hybrid Encoder. The Dual-Domain Hybrid Encoder performs unified encoding of the frequency domain representations and the temporal representation of dance sequences. The output of the unified encoding is then used as the keys and values in the cross-attention layer of the dance decoder.

## 4 EXPERIMENTS

### 4.1 EXPERIMENT SETUP

**Dataset.** In our experiments, we tested our method using the open source dataset AIST++ (Li et al., 2021), which contains paired music and dance sequences. The AIST++ dataset contains 1,408 dance sequences recorded in joint rotation and root translation representations. The dance sequences in the AIST++ dataset vary in length, ranging from 7.4 seconds to 48 seconds.

**Implementation details.** In our experiments, the length of both motion representations and music features is 128 frames, with a frame rate of 30 frames per second, corresponding to a duration of approximately 4.3 seconds. We used the Adan optimizer during training and set the learning rate to 1e-4. The training of the model consists of two stages. Prior to training the dance generation framework, we first train the frequency domain feature extractor with the same data used for the

dance generation framework. Once the frequency domain feature extractor is trained, we freeze its weights and then train the dance generation framework. During this training stage, we employ an exponential moving average (EMA)(Klinker, 2011) strategy. In the second stage of training, our model has 64.5M trainable parameters and was trained on four NVIDIA 3090 GPUs for 2000 epochs, with a batch size of 32.

## 4.2 RESULTS

We compare our method with FACT(Li et al., 2021), Bailando(Siyao et al., 2022), EDGE(Tseng et al., 2023), and BADM(Zhang et al., 2024a). FACT is an sequence-based dance generation model based on a transformer model; Bailando is also a sequence-based dance generation model that combines a VQ-VAE model with a transformer model; EDGE is a diffusion dance generation framework utilizing a transformer-based model; and we also compared with BADM, which uses a transformer-based diffusion framework similar to EDGE, incorporating both preceding and subsequent sequence information into the generation process by segmenting the motion sequence. Furthermore, we evaluated the quality of dance generation by our method and selected methods under conditions of fast-paced and slow-paced music. We used librosa (Mcfee et al., 2015) to calculate the BPM(Beat Per Minute) values of the music in the AIST++ test set and divided the test set into two equally-sized subsets based on these BPM values. The corresponding results are presented in Table 2.

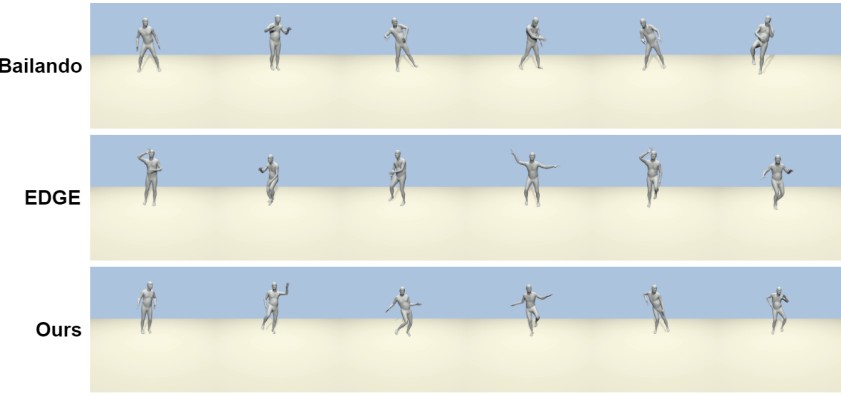

Figure 4: Visualization of generated dance sequence. Our method is capable of generating dance sequences that are more expressive and feature more natural and diverse transitions.

**Physical plausibility.** We adopt the Physical Foot Contact(PFC) score(Tseng et al., 2023) as one of the metrics to measure the quality of the dance sequences. The motivation behind the PFC score is that high-quality dances should possess good physical plausibility. The PFC score is based on the observation that, in dances, the acceleration of the dancer's center of mass must be supported by contact with the ground through either one or both feet. Because dance movements differ from general human movements and often involve foot sliding, which can be an element of choreography, the PFC score provides a better evaluation of the physical plausibility of dance sequences compared to simply calculating foot sliding related metrics, and reflects the quality of the generated dance sequences. As shown in the Table 1, HyDance achieves an improvement of 0.278 over EDGE in the PFC metric, and shows improvements of 0.996, 0.496, and 0.166 respectively over FACT, Bailando, and BADM. This demonstrates that HyDance produces dance sequences with better motion quality.

**Beat alignment score.** We follow(Siyao et al., 2022) to compute the Beat Alignment Score(BAS) to measure the alignment of the generated dance sequences with the beat of the music. The BAS score reflects the temporal distance between dance beats and the beats of the conditioning music, where dance beats are defined by the local minima of joint velocities.

**Diversity.** Following (Siyao et al., 2022; Li et al., 2021), we computed the diversity metric, which quantifies the average feature distance of the generated dance sequences in both kinetic ($DIV_k$) and geometric ($DIV_g$) feature spaces. To mitigate the impact of irregular jittering present in some generated dance results that can inflate diversity scores, we adopted the approach from (Tseng et al., 2023), taking results with DIV scores that are closer to the ground truth as being superior.

Table 1: Compare with SOTAs on AIST++ test set. ↓ means lower is better, ↑ means higher is better, and → means closer to the ground truth is better.

| Method | PFC↓ | Beat Align Score↑ | $\text{DIV}_k \rightarrow$ | $\text{DIV}_g \rightarrow$ | Ours Win Rate |
|---|---|---|---|---|---|
| Ground Truth | 1.332 | 0.2463 | 9.41 | 7.28 | 48.81% |
| FACT | 2.254 | 0.2209 | 10.85 | 6.14 | 75.00% |
| Bailando | 1.754 | 0.2332 | 7.92 | **7.72** | 73.80% |
| EDGE | 1.536 | 0.2281 | **9.48** | 5.72 | 72.61% |
| BADM | 1.424 | 0.2366 | 8.29 | 6.76 | 62.50% |
| HyDance(Ours) | **1.258** | **0.2710** | 7.61 | 5.30 | N/A |

Table 2: Quantitative evaluation on subsets of AIST++ test set with different BPMs.

| Method | Low BPMs | | | | |
|---|---|---|---|---|---|
| | PFC↓ | BAS↑ | $\text{DIV}_k \rightarrow$ | $\text{DIV}_g \rightarrow$ | Ours Win Rate |
| Ground Truth | 1.243 | 0.2641 | 7.31 | 7.57 | 50.00% |
| Bailando | 1.721 | 0.2238 | **7.28** | **6.64** | 88.10% |
| EDGE | 1.524 | 0.2566 | 8.32 | 5.82 | 78.57% |
| HyDance(Ours) | **1.102** | **0.2625** | 7.96 | 5.26 | N/A |

| Method | High BPMs | | | | |
|---|---|---|---|---|---|
| | PFC↓ | BAS↑ | $\text{DIV}_k \rightarrow$ | $\text{DIV}_g \rightarrow$ | Ours Win Rate |
| Ground Truth | 1.524 | 0.2138 | 8.42 | 6.40 | 47.62% |
| Bailando | 1.851 | 0.2471 | 7.32 | **6.43** | 64.28% |
| EDGE | 1.764 | 0.2240 | **8.62** | 4.62 | 69.05% |
| HyDance(Ours) | **1.545** | **0.2776** | 8.06 | 5.65 | N/A |

**User study.** Human audience appreciation is a key aspect when evaluating dance sequences. To better evaluate the dance generation results of our method, we conducted a user study to qualitatively compare the dance quality generated by our method with that of other methods. The study involved 14 participants, each of whom watched 24 pairs of randomly picked videos of generated dances, with each pair containing a dance generated by our method and another dance generated by a different method. Participants were asked to select the video that "have higher quality dance movements, more akin to dances performed by human dancers", or to choose the option indicating that the quality of the dances was too close to determine a difference. We calculated the win rate of our method relative to others based on the user study results, which are presented in Table 1.

**frequency domain transformation analysis.** We further illustrate the impact of high-frequency information in dance sequences on the perception of dance movements by performing frequency domain transformation analysis on the generated dance sequences. Through our observations of previous methods' dance generation results, we noted that the movements produced by these methods tend to be overly smooth. This issue becomes more evident when using the generated dance sequences to drive more realistic human models such as SMPL(Loper et al., 2015). We compare the frequency domain transformation results of dance sequences generated by our method with those

Table 3: Comparison of our method with SOTAs in terms of FID metrics.

| Method | $\text{FID}_k \downarrow$ | $\text{FID}_g \downarrow$ |
|---|---|---|
| Ground Truth | 17.10 | 10.60 |
| FACT | 35.35 | 22.11 |
| Bailando | 28.16 | 9.62 |
| EDGE | 42.16 | 22.12 |
| HyDance(Ours) | 58.53 | 21.25 |

generated by EDGE, and find that the energy of high-frequency components is usually lower, as shown in Fig.5. This indicates that the high-frequency features in the dance sequences are not well preserved, leading to dance movements that appear overly smooth and lack expressiveness. To illustrate this issue more intuitively, we applied a low-pass filter to the dance sequences generated by our method and the ground truth dance sequences, then rendered them and compared the visual presentation before and after the filtering operation. Please refer to the video in the supplementary material for this comparison.

**FID results.** Fréchet Inception Distance(FID) is a widely used metric for the overall evaluation of generative models. FID quantifies the difference between the empirical distribution of the generated results and the ground truth distribution. Previous works(Li et al., 2021; Siyao et al., 2022) on dance generation use the FID metric to assess the quality of generated dance sequences. FACT(Li et al., 2021) proposed extracting different motion features from dance sequences to compute $\text{FID}_k$ and $\text{FID}_g$ metrics. However, EDGE(Tseng et al., 2023) pointed out that the reliability of this metric for assessing dance sequence generation quality is questionable. Through experiments, EDGE demonstrated that although some generated dance sequences achieved the best performance in the $\text{FID}_g$ metric, they received pretty low ratings in user evaluations, suggesting that this metric may not reliably reflect the quality of dance generation results on the AIST++ dataset. For the $\text{FID}_k$ metric, EDGE found through evaluations of models trained for different durations that, although the user evaluation performance of the dance generation results gradually improved, the $\text{FID}_k$ metric did not show a similar trend. Therefore, the FID metric may not reliably reflect the quality of dance generation results on the AIST++ dataset. We present the comparison results of our method with other methods in terms of the FID metric in Table 3.

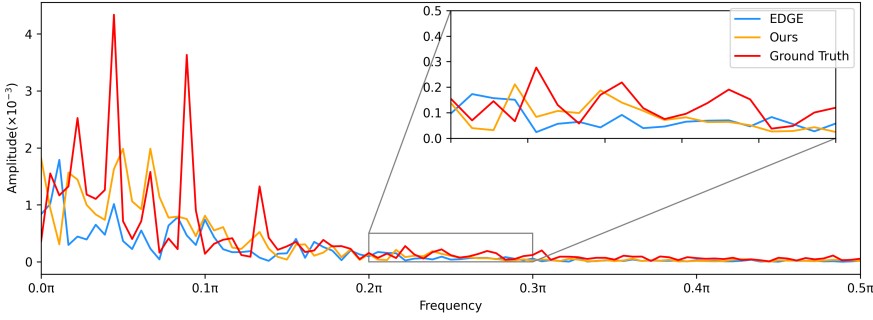

Figure 5: Frequency spectrum plot of generated dance sequences. We compare the frequency domain analysis results of dance sequences generated by our method and EDGE. Notice that the amplitudes of higher frequencies are generally lower in the generation results of EDGE, resulting in overly smooth and less detailed dance sequences. This aligns with our assumption that high-frequency features in dance sequences influence the expressiveness of the final dance performance.

Table 4: Ablation study results.

| Method | PFC↓ | Beat Align Score↑ | $\text{DIV}_k \rightarrow$ | $\text{DIV}_g \rightarrow$ | Ours Win Rate |
|---|---|---|---|---|---|
| Ground Truth | 1.332 | 0.2463 | 9.41 | 7.28 | |
| HyDance | 1.258 | 0.2710 | 7.61 | 5.30 | |
| w/o Freq.-Domain representaions | 2.108 | 0.2395 | 4.01 | 3.99 | 78.57% |
| w/o Dual-Domain Hybrid Encoder | 1.314 | 0.2608 | 4.81 | 5.29 | 61.90% |

### 4.3 ABLATION STUDY

In this section, we perform ablation studies to validate the the following components: (1). frequency domain motion representations, (2). the Dual-Domain Hybrid Encoder.

**frequency domain motion representations.** In this part of experiments, we removed the frequency domain motion representations from the input to the model and accordingly removed the frequency domain feature extractor and frequency domain encoder from the dance generation framework. The results after removing these components are shown in the second row of Table 4. The PFC score significantly increased, indicating a substantial decrease in the physical plausibility of the generated dance sequences. Additionally, the BAS score also dropped considerably, indicating poorer alignment of the dance sequences with the music beats. Moreover, the $\text{DIV}_k$ and $\text{DIV}_g$ metrics also showed a significant decrease, indicating a decline in the diversity of the generated dance sequences.These experiments demonstrate that the frequency domain dance representation helps improve the quality of the generated dance sequences as well as spatial-temporal alignment.

**Dual-Domain Hybrid Encoder.** The Dual-Domain Hybrid Encoder helps the dance generation framework utilize the frequency domain motion representation more effectively. As shown in Table 4, when we removed this component from the framework, the performance of the generated dance sequences slightly degraded in terms of physical plausibility and beat alignment, indicating that the model can improve the quality of generated dance sequences through the Dual-Domain Hybrid Encoder.

## 5 CONCLUSION

In this paper, we propose HyDance. We argue that the high-frequency features in dance sequences contain numerous motion details, which are important for determining whether the final dance performance is close to that of a human dancer. We find that previous works focus only on the temporal features of dance, neglecting the dynamic features in the frequency domain, which leads to generated dances lacking dynamic performance and expressiveness. Therefore, we combine both frequency domain and temporal features to generate dance movements. To fuse these features more effectively, we introduce the HyDance network. Our method is validated on public datasets, achieving state-of-the-art performance, and ablation experiments further demonstrate the effectiveness of the proposed modules.

**Limitations and future work.** Although our method generates dance sequences with improved expressiveness and details, we also note in our experiments that the root node displacement trajectories in the dance sequences generated by our method are not adequately represented. We believe that the possible reason for this issue is that the frequency-domain feature extractor in the framework focuses more on local periodic features of the dances, leading to poor capture of the root node displacement characteristics. Therefore, a meaningful direction for future work could be to explore better ways to encode dance sequences, ensuring that both global displacement features and local dance movement features are well captured.

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
