# OpenReview forum: "HyDance: A Novel Hybrid  Dance Generation Network with temporal and  frequency features"
_ICLR.cc/2025/Conference — Submitted to ICLR 2025_

### Official Review · Reviewer_vNB2 · 2024-10-30

**Soundness:** 2
**Presentation:** 2
**Contribution:** 1
**Rating:** 5
**Confidence:** 4

**Summary:**

This paper introduces HyDance, a novel diffusion-based model for generating dance from music. The core of the proposed approach lies in its ability to encode motion features in both the temporal and frequency domains, allowing for a complementary interaction during dance generation. Empirical evaluations on the AIST++ datasets demonstrate that HyDance surpasses existing state-of-the-art methods both quantitatively and qualitatively.

**Strengths:**

- The idea of leveraging frequency domain motion features to enhance music-driven dance generation is promising.
- The paper is written clearly and straight-forward to follow. Demo video in the supplementary material is also helpful.

**Weaknesses:**

- Overall contribution is limited. The frequency domain motion feature extractor was proposed in PAE paper. This paper simply adapts it to music-to-dance generation. Considering that the original PAE paper already worked on motion sequences, the adaptation effort is also minor even for an application paper. Besides, the scope of taking the advantage of frequency domain motion features should not be limited to dance generation. It will be still useful for any conditional (complex) motion sequence generation problems but unfortunately authors did not investigate more.
- Experiments are conducted on AIST++ only. While it is a popular dataset, it has very limited music pieces, which means the model will definitely overfit the music data in AIST++. It would make more sense to test how well it can generalize to more general music pieces.
- More analyses are required for frequency-related performance. For example, AIST++ has different dance motions for high and low BPMs. Is there any performance gap across different BPM?
- Some implementation details are missing such as the number of trainable parameters. How about the inference speed? With these additional hybrid encoders, will the generation speed be slower?
- Definitions and analyses of $DIV_k$ and $DIV_g$ are missing.
- Details of user study are also missing. How did you select the video pairs? Why not have an additional 'neutral' option? How do you make sure 14 video pairs are sufficient?
- The Figure 4 is not really informative. Examples in the demo video look better. Maybe a spectrogram (temporal-frequency) magnitude visualization could help?

**Questions:**

See questions in weaknesses.

---

> ### Author Response · Authors · 2024-11-21
>
> Reviewer **vNB2**,
>
> Thank you for reviewing our paper and your valuable suggestions. We have carefully considered your feedback and provide our responses below:
>
> **Weakness:**
>
> **1. Overall contribution is limited.**
>
> We thank the reviewer for the insightful suggestions, which have effectively helped us to improve our work.
>
> a. Overall contribution
>
> Our motivation is to address the issue of models generating overly monotonous dance movements,  rather than focusing on the application of PAE. Our contribution is not an exploration of the application of PAE. Instead, we propose a framework that integrates frequency-domain and time-domain features for dance generation, and we have conducted experimental tests to validate this framework.
>
> In our experiments, we observed that the more compact frequency-domain feature space indeed helps the model better capture motion transitions within dance sequences. However, using only frequency-domain representations does not yield satisfactory dance movement results, especially concerning root node movement and limb positioning. Therefore, we designed a generation framework that leverages the complementary strengths of both representations.
>
> b.  The scope of taking the advantage of frequency domain motion features should not be limited to dance generation.
>
> This is an excellent idea. However, our current work focuses on generating high-quality, expressive dance motions. We plan to open-source our code to support further research in complex motion generation.
>
> **2. Generalization ability.**
>
> We agree with the reviewer that generalization ability is a critical aspect of dance motion generation models. To demonstrate the generalization capability of our method, we compared the dance generation results of our approach with those of the reference method under randomly selected in-the-wild music conditions. We have uploaded the result videos to a fully anonymous repository, which can be accessed via the following link:
> https://github.com/null770/submission5560/blob/main/inthewildtest.mp4
>
> From the generated result videos, our method demonstrates decent performance under in-the-wild music conditions of varying styles. It produces expressive and rhythmically aligned dance sequences, indicating the good generalization capability of our approach.
>
> **3. More analyses for frequency-related performance.**
>
> We appreciate the reviewer’s novel and constructive suggestions. We have added comparisons of dance generation results under different BPM music conditions in the revised version of the paper (Section 4.2, Table 2, Line 393). The revisions can also be found in the table below:
>
> ### Table 2. Quantitative evaluation on subsets of AIST++ test set with different BPMs
>
> **Low BPMs**
> | **Method**      | **PFC↓** | **BAS↑**  | **DIV_k→** | **DIV_g→** | **Ours Win Rate** |
> |------------------|----------|-----------|------------|------------|--------------------|
> | Ground Truth     | 1.243    | 0.2641    | 7.31       | 7.57       | 50.00%            |
> | **Bailando**     | 1.721    | 0.2238    | **7.28**   | **6.64**   | 88.10%            |
> | **EDGE**         | 1.524    | 0.2566    | 8.32       | 5.82       | 78.57%            |
> | **HyDance(Ours)**| **1.102**| **0.2625**| 7.96       | 5.26       | N/A               |
>
> ---
>
> **High BPMs**
> | **Method**      | **PFC↓** | **BAS↑**  | **DIV_k→** | **DIV_g→** | **Ours Win Rate** |
> |------------------|----------|-----------|------------|------------|--------------------|
> | Ground Truth     | 1.524    | 0.2138    | 8.42       | 6.40       | 47.62%            |
> | **Bailando**     | 1.851    | 0.2471    | 7.32       | **6.43**   | 64.28%            |
> | **EDGE**         | 1.764    | 0.2240    | **8.62**   | 4.62       | 69.05%            |
> | **HyDance(Ours)**| **1.545**| **0.2776**| 8.06       | 5.65       | N/A               |
>
> **4. Implementation details**
>
> In the revised paper, we have included a description of trainable parameters. Regarding inference speed, it is highly dependent on the GPU and memory size, making it challenging to provide a specific description. For reference, generating a 15-second dance sequence on an NVIDIA 3090 GPU takes approximately 2 to 3 seconds.
>
> **5. Definitions and analyses of DIV metric.**
>
> Thank you for pointing out the shortcomings. We have refined the descriptions of the evaluation metrics in the revised paper(Section 4.2, Lines 374-377).
>
> **6. Details of user study.**
>
> We are grateful for the reviewer’s constructive feedback. We re-conducted subjective evaluation experiments and enhanced the descriptions of the experiments.(Section 4.2, Lines 417-425)
>
> **7. The Figure 4 is not really informative.**
>
> In Figure 4(Section 4.2, Line 469), we have added ground truth data to facilitate better understanding.
>
> We sincerely appreciate the reviewer’s insightful suggestions, which have been instrumental in helping us improve this work.

---

> > ### Comment · Reviewer_vNB2 · 2024-11-25
> >
> > Thanks authors' efforts for new experimental results and clarifications. While I acknowledge the proposed hybrid method seems helpful for dance generation according to good quality generated samples, I still think the overall improvement is relatively incremental and some analyses are not convincing. For example, I couldn't tell if the "Ours" is more aligned with the "Ground Truth" than "EDGE" shown in the updated Figure 5 even though "Ours" have more peaks than "EDGE". Besides, could you explain why DIV metrics are not as good as others? It is fine that some quantitative results are not the best but authors should provide more insights for failures. Based on the rebuttal, I will raise my score but still towards borderline reject.

---

> > > ### Author Response · Authors · 2024-11-27
> > >
> > > Thank you for your response and for improving our score. Regarding your concerns, we provide the following replies:
> > >
> > > 1. **About Figure 5**
> > >
> > >     We sincerely apologize for any confusion caused. The purpose of including Fig 5. was to demonstrate, in combination with the videos provided in the supplementary materials, that the presence of high-frequency components in the motion sequence can influence the quality of the generated dance movements.
> > >
> > > 2. **Why DIV metrics are not as good as others, and more insights into failures should be provided**
> > >
> > >     We appreciate the reviewer’s feedback and constructive suggestions. The performance of the DIV metrics may be influenced by multiple factors. For example, we observed that when the root node's movement is more pronounced in the motion sequence, the DIV$_k$metric tends to perform significantly better. However, a more detailed analysis of the performance of the DIV metric would require additional experimental evidence. Due to time constraints, we are unable to conduct and provide such in-depth analyses at this stage. Nonetheless, we are very grateful for the reviewer’s insightful suggestions.

---

### Official Review · Reviewer_wujW · 2024-11-01

**Soundness:** 3
**Presentation:** 3
**Contribution:** 2
**Rating:** 6
**Confidence:** 5

**Summary:**

The HyDance paper proposes a novel method for music-driven dance generation using a transformer-based diffusion network that incorporates both temporal and frequency domain representations. The authors emphasize the limitations of prior works that only leverage temporal representations, leading to oversmoothed, less dynamic dance sequences. By integrating frequency domain features, HyDance reportedly generates more expressive, realistic dances aligned with musical beats.

**Strengths:**

1. The approach does present some novel aspects in its methodolgy:
    i). Hybrid representation: Integrating both temporal and frequency representations for better capturing dance dynamics.
    ii). Dual-Domain Hybrid Encoder: This component introduces an interesting method to combine temporal and frequency-based motion representations, which is less common in dance generation tasks.

2. The experiments are generally well-structured, with comparisons against state-of-the-art methods like FACT, Bailando, EDGE, and BADM on the AIST++ dataset.

**Weaknesses:**

1. Qualitative results demonstration: Visual examples comparing generated sequences with other SOTA methods would be more helpful to show that the proposed method indeed generate better dynamics.

2. w/o Dual-Domain Hybrid Encoder, the model seems to achieve comparable performance against the full model except the DIV_k metrics. Could a human study conducted on these ablation versions to show that without w/o Dual-Domain Hybrid Encoder, the model cannot generate expressive dance motions.

**Questions:**

See weaknesses above.

---

> ### Author Response · Authors · 2024-11-21
>
> Reviewer **wujW**,
>
> Thank you very much for your thorough review of our paper and the valuable suggestions for improvement.
>
> **Weaknesses:**
>
> **1. Qualitative results demonstration:**
>
> We agree with the reviewer's perspective. We have included comparison figures of the dance rendering results alongside those from other methods.(Section 4.2, Figure 4, Line 342)
>
> **2. w/o Dual-Domain Hybrid Encoder, the model seems to achieve comparable performance against the full model except the DIV_k metrics.**
>
> Thank you for your constructive suggestion. We have re-conducted the subjective evaluation experiment and added subjective result comparisons for the ablation studies. (Section 4.2, Table 4, Line 487) The revisions can also be found in the table below:
> ### Table 4. Ablation study results
>
> | **Method**                                | **PFC↓** | **Beat Align Score↑** | **DIV_k→** | **DIV_g→** | **Ours Win Rate** |
> |-------------------------------------------|----------|------------------------|------------|------------|--------------------|
> | **Ground Truth**                          | 1.332    | 0.2463                | 9.41       | 7.28       | -                  |
> | **HyDance**                               | 1.258    | 0.2710                | 7.61       | 5.30       | -                  |
> | **w/o Freq.-Domain representations**      | 2.108    | 0.2395                | 4.01       | 3.99       | 78.57%            |
> | **w/o Dual-Domain Hybrid Encoder**        | 1.314    | 0.2608                | 4.81       | 5.29       | 61.90%            |
>
> We greatly appreciate the reviewers’ insightful suggestions, which have helped us enhance the quality of the manuscript.

---

> > ### Author Response · Authors · 2024-11-27
> >
> > Reviewer **wujW**,
> >
> > We hope our earlier response has addressed your concerns. If there are any additional points or suggestions you would like us to consider, please feel free to let us know. Thank you again for your valuable feedback!

---

### Official Review · Reviewer_LHkM · 2024-11-02

**Soundness:** 3
**Presentation:** 3
**Contribution:** 2
**Rating:** 6
**Confidence:** 4

**Summary:**

The paper presents a method for generating 3D pose sequences of dances from input audio. The authors consider both the time and frequency domain representations of dance motions, and propose a Dual-Domain Hybrid Encoder to combine the temporal and frequency information, particularly the higher-frequency information that can get suppressed in traditional attention mechanisms. They leverage this combined representation in a transformer-based diffusion network to generate dances with high-frequency movements. They show the benefits of their proposed approach through quantitative and qualitative comparisons, ablation experiments, and a user study.

**Strengths:**

1. The idea of explicitly featurizing high-frequency information in the encoding process is intuitive and well-motivated, and the technical aspect of preserving those in the synthesized outputs is soundly presented.

2. The experimental results highlight the benefits of the proposed approach. Particularly, the visual results clearly show the benefits of leveraging high-frequency information for dances.

**Weaknesses:**

1. The authors motivate the utility of capturing high-frequency information at transitions of dance movements, which are arguably in sync with the music beats. While this is an empirically plausible idea, it lacks any discussion with similar ideas explored differently in the existing literature, such as [A] (not cited in the paper), which separately generates higher-frequency beat poses and lower-frequency in-between poses. Some discussions with other approaches exploring a similar idea would help contextualize the paper in the literature.

[A] Bhattacharya, Aneesh, Manas Paranjape, Uttaran Bhattacharya, and Aniket Bera. "DanceAnyWay: Synthesizing Beat-Guided 3D Dances with Randomized Temporal Contrastive Learning." In Proceedings of the AAAI Conference on Artificial Intelligence, vol. 38, no. 2, pp. 783-791. 2024.

2. The key contribution of utilizing frequency-domain information can be explained in more depth. It would be informative to understand how the frequency domain information impacts the generative performance for different types of dances (e.g., slow-moving dances like waltz vs. dances with rapid movements like hip-hop). Also, since the frequency domain representation already captures information on the entire frequency spectrum, what additional information does the time-domain representation provide? Are there any particular correlations of different MFCC or Chroma components in the audio with the frequency representations of the different joints, particularly as the highest frequency in the dance increases (that is, the dance becomes progressively faster-paced)?

3. In addition, the quantitative and ablation experiments can also be explained in more detail to highlight the proposed contributions better. For example, why do the diversity scores (particularly DIV_k) drop by nearly half when the frequency representations and the Dual-Domain Encoder are removed (Table 3)? How are the generated dances able to achieve better scores than the Ground Truth on various metrics (Tables 2 and 3)?

4. Some details on the user study are also missing. Did the authors allow for ties in the study? Otherwise, the win rates might be inflated even if the generated dances are suboptimal. Further, the win rate over ground-truth dances is above 50%, which, coupled with poorer quantitative numbers of the ground-truth, raises further questions on what the ground-truth actually looks like and how the training process generates results that supposedly surpass the ground-truth in quality.

**Questions:**

1. How is the contact loss term (Eqn. 7) penalized when the predicted foot contact, $\hat{b}^{(i)}$, is wrong?

2. Please show the y-axis (amplitude) labels in Fig. 4 to make the figure readable. Currently, it is hard to understand the scale of improvements in the high-frequency region.

3. For the ablation experiment without the Dual-Domain Encoder, what encoder is actually being used?

---

> ### Author Response · Authors · 2024-11-21
>
> Reviewer **LHkM**,
>
> Thank you very much for your thorough review of our paper and the valuable suggestions for improvement. We have carefully considered your feedback, and here is our response:
>
> **Weaknesses:**
>
> **1. Discussions with other approaches exploring a similar idea**
>
> We agree with the reviewer and have added a discussion of related work in the revised manuscript (Section 2.2, Lines 160-161).
>
> **2. The key contribution of utilizing frequency-domain information can be explained in more depth.**
>
> Thank you for your constructive suggestions. We have included comparisons of results under different tempos of music in the updated version of the paper (Section 4.2, Table 2, Line 392).  The revisions can also be found in the table below:
> ### Table 2. Quantitative evaluation on subsets of AIST++ test set with different BPMs
>
> **Low BPMs**
> | **Method**      | **PFC↓** | **BAS↑**  | **DIV_k→** | **DIV_g→** | **Ours Win Rate** |
> |------------------|----------|-----------|------------|------------|--------------------|
> | Ground Truth     | 1.243    | 0.2641    | 7.31       | 7.57       | 50.00%            |
> | **Bailando**     | 1.721    | 0.2238    | **7.28**   | **6.64**   | 88.10%            |
> | **EDGE**         | 1.524    | 0.2566    | 8.32       | 5.82       | 78.57%            |
> | **HyDance(Ours)**| **1.102**| **0.2625**| 7.96       | 5.26       | N/A               |
>
> ---
>
> **High BPMs**
> | **Method**      | **PFC↓** | **BAS↑**  | **DIV_k→** | **DIV_g→** | **Ours Win Rate** |
> |------------------|----------|-----------|------------|------------|--------------------|
> | Ground Truth     | 1.524    | 0.2138    | 8.42       | 6.40       | 47.62%            |
> | **Bailando**     | 1.851    | 0.2471    | 7.32       | **6.43**   | 64.28%            |
> | **EDGE**         | 1.764    | 0.2240    | **8.62**   | 4.62       | 69.05%            |
> | **HyDance(Ours)**| **1.545**| **0.2776**| 8.06       | 5.65       | N/A               |
>
> We aimed to combine time-domain and frequency-domain representations to address the issue of overly smooth dance movements in generation networks. The more compact frequency-domain feature space allows the model to better capture motion transitions, resulting in dance movements with increased motion transitions.
>
> However, during experiments, we also observed that frequency-domain representations alone were insufficient for achieving satisfactory dance movement results, particularly in root node movement and limb positioning. Thus, we designed the model to combine the strengths of both representations.
>
> **3. Experiments can also be explained in more detail.**
>
> Thank you for your insightful suggestions for improving this paper. We have provided an explanation for the DIV metric in the revised paper(Section 4.2, Lines 374-377).
> Regarding instances where certain metrics exceed the ground truth, this is due to the calculation principles of these metrics. For example, the PFC score tends to be lower if dance sequences have less movement of the dancer’s root node, which is also reflected in our evaluation of dance results generated under different tempos of music (Table 2 Above). As for the BAS metric, since it measures the distance between music beats and dance beats, the integration of frequency-domain representation helps our method generate dance sequences with more reasonable movement transitions, leading to potentially higher BAS scores.
>
> **4. Some details on the user study are also missing.**
>
> We have re-conducted the subjective evaluation experiment, increased the number of dance sequences for comparison, and introduced a neutral option. These changes are reflected in the revised version of the paper(Line 379, Table 1 and Line 392, Table 2).
>
> **Questions:**
>
> **1.About the contact loss term**
>
>  The contact loss term originates from EDGE [A], aiming to encourage the model to predict contact labels and ensure that the generated movements are consistent with its own predicted labels. When the model predicts foot contact with the ground but the foot velocity remains high, this loss increases.
>
> **2.About freq. analysis figure**
>
>  We have added Ground Truth data to Figure 4 in the revised manuscript (Line 469) to show the difference and improvement.
>
> **3.Details of user study**
>
>  In the ablation study of the hybrid encoder, we do not use other encoders. Instead, we directly input the results from the frequency-domain and time-domain representation encoders into the Dance Decoder as the query and key, respectively.
>
> We are grateful for the time and effort you have invested in reviewing our work. Your suggestions have been instrumental in improving our paper. We hope that our responses are satisfactory, and if possible, please consider increasing the score for this submission.
>
> [A] Tseng, J., Castellon, R. and Liu, K., 2023. Edge: Editable dance generation from music. In *Proceedings of the IEEE/CVF Conference on Computer Vision and Pattern Recognition* (pp. 448-458).

---

> > ### Comment · Reviewer_LHkM · 2024-11-25
> > **Thanks for the rebuttal**
> >
> > I thank the authors for their detailed rebuttal, which addresses my concerns. Overall, I maintain my original acceptance recommendation on account of the soundness of the frequency-based approach and the promising results. However, apologies if my earlier comment was unclear - in the current Fig. 5, the y-axis (amplitude) values are still missing. Without knowing the amplitude values in the plot (especially in the inset zooming in on the high-frequency range), it is hard to know how much of an amplitude difference there actually is between the methods.

---

> > > ### Author Response · Authors · 2024-11-27
> > >
> > > Thank you for your active engagement in the discussion. We have revised Fig 5. in the updated version of the paper. If you have any other concerns, please do not hesitate to let us know!

---

> > > > ### Comment · Reviewer_LHkM · 2024-11-27
> > > > **Figure looks good**
> > > >
> > > > Thank you. The figure is now readable, and I can understand the differences between the methods.

---

### Official Review · Reviewer_YEBL · 2024-11-04

**Soundness:** 3
**Presentation:** 2
**Contribution:** 3
**Rating:** 5
**Confidence:** 4

**Summary:**

This paper proposes a method to address the issue of unnatural movement generation in music-driven dance generation tasks by utilizing the frequency domain characteristics of movements to supplement the missing information in temporal features. The overall framework is based on the Diffusion model. Additionally, an encoder that integrates frequency domain features with temporal features has been designed to implement the proposed method of feature fusion for motion generation. The experiments are sufficient, and the maniscript is of high quality.

**Strengths:**

This paper introduces a method that enhances the naturalness of dance movements to align with human aesthetics by leveraging the frequency domain characteristics of movements. It employs a frequency domain feature extractor to capture the frequency domain features of the movements and designs a corresponding feature fusion encoder that combines temporal features, thereby more effectively improving the quality of generated movements.
This paper provides a detailed explanation of the proposed method, conducts a variety of experiments, and analyses the results. Particularly, it performs ablation studies on the designed frequency domain feature extractor and feature fusion encoder to demonstrate the effectiveness of these two modules. Additionally, the paper also designs a user study to validate the proposed method.
This paper provides an detailed description of the proposed method and its structure is well-organized. The experimental results are presented in a visual manner using charts and graphs.

**Weaknesses:**

1. The Section 3.1 of the paper regarding the representation of dance movement contains some ambiguity. Specifically, the description of "contact label" is unclear. Tseng et al. describe it as "the heels and toes of the left and right feet", but this paper's description of "front and back" is not clear.
2. In Section 3.1, concerning the "a music sequence of length $L$", the unit of $L$ should be specified as "frames".
3. In Figure 3, the processing of "music, time tokens" does not align with the explanation provided in Section 3.4 of the paper.
4. Figure 4 could incorporate the spectrum of the Ground Truth.
5. Some parts are not very clear in this version. e.g.,
    Enlarge the font in Figure 1.
    Provide additional explanations for the arrows in Tables 1, 2, and 3.
    Offer supplementary explanations for the evaluation metrics "$DIV_k$" and "$DIV_g$".

**Questions:**

1. In Section 3.3, regarding the explanation of “L_f2m”, there is a distinction that needs clarification between “the frequency domain representation of the motion sequence d^f” and “the reconstructed temporal representation of the dance sequence d ̂^f” compared to the “d_f” mentioned in Section 3.1. What is the difference between these representations?
2.In Figure 2, why does d^f input into the “Temporal Encoder”? What is the meaning of the arrows from the “Temporal Encoder” and the “Freq Domain Encoder” modules to the “Dance Decoder”?

---

> ### Author Response · Authors · 2024-11-21
>
> Reviewer **YEBL**,
>
> Thank you very much for your thorough review of our paper and for the valuable suggestions. We have carefully considered your comments, and here is our response:
>
> **Weakness:**
>
> **1. & 2. Description of representations**
>
> We agree with the reviewer that the descriptions of the contact label and the duration of the music could be more detailed.  The foot contact label is the heels and toes of the left and right feet the same with  previous work EDGE[A].  “a music sequence of length L” wil be modified to ““a music sequence of  L frames”. Accordingly, we have made the necessary revisions in the revised version of the paper (Section 3.1, Line 192 and Lines 198-199).
>
> **3. Misalignment between the figure and description**
>
> We apologize for the incorrect descriptions. The confusion in the description has been  corrected in the revised version of the paper  (Section 3.4, Lines 287-288). The relevant description has been revised to: “Finally, the noisy motion sequences in two different representations, are fed into the dance decoder together with the unified motion feature sequences obtained from the hybrid encoder”
>
> **4. Figure 4 could incorporate the spectrum of the Ground Truth.**
>
> We agree with the reviewer and have added Ground Truth data to the spectrogram in the revised version of the paper (Figure 4, Line 469).
>
> **5. Some parts are not very clear**
>
> The figures in the paper have been revised, and we have provided explanations for the meanings of the arrows and the DIV metrics (Section 4.2, Lines 374-377, Lines 379-380).
>
> **Questions:**
>
> **1. Clarification between notations**
>
> The notation $d_f$ in Section 3.1 is a typo and should be corrected to $d^f$, consistent with the notation in Section 3.3, where it represents the sequence of dance in the frequency domain representation. $\hat{d^f}$ denotes the dance sequences in the temporal domain representation that is obtained by using $d^f$ as the query input to the Dance Decoder.
>
> **2. In Figure 2, why does d^f input into the “Temporal Encoder”?**
>
> In Figure 2, we mistakenly depicted the two encoders. We apologize for the confusion caused by it, This has been corrected in the revised version of the manuscript.
>
> **3. What is the meaning of the arrows from the “Temporal Encoder” and the “Freq Domain Encoder” modules to the “Dance Decoder”?**
>
> The arrows from the “Temporal Encoder” and the “Freq Domain Encoder” modules to the “Dance Decoder” indicate that the outputs at the tails of the arrows serve as the query inputs to the Dance Decoder.
>
>
> Once again, we express our sincere gratitude for the time you have invested in reviewing our work. Your feedback have significantly contributed to the enhancement of our paper. If our revisions and responses meet with your expectations, please consider increasing the score.
>
> [A] Tseng, J., Castellon, R. and Liu, K., 2023. Edge: Editable dance generation from music. In *Proceedings of the IEEE/CVF Conference on Computer Vision and Pattern Recognition* (pp. 448-458).

---

> ### Author Response · Authors · 2024-11-27
>
> Reviewer **YEBL**,
>
> We hope our previous response addressed your concerns. If there are any additional points or suggestions you'd like us to address, please feel free to let us know. Thank you again for your valuable feedback!

---

### Author Response · Authors · 2024-11-22

We sincerely appreciate the time and effort that all reviewers have dedicated to evaluating our work. We have greatly benefited from the constructive feedback provided. We have been greatly encouraged by the reviewers’ recognition of aspects of our work such as **intuitive, well-motivated and promising idea(LHkM, vNB2)**, **well organized description of proposed method(YEBL, wujW)**, **methodology with some novel aspect(wujW)**, etc. We apologize for any misunderstandings or inconvenience caused by our negligence and omissions in our initial manuscript. In the revised version, we have carefully considered and incorporated the reviewers’ suggestions, and we provide detailed responses to the concerns below.

---

### Author Response · Authors · 2024-11-29
**Dear reviewers**

Dear reviewers,

We sincerely thank all the reviewers for their diligent effort. We have carefully considered the suggestions and concerns raised and have accordingly added experiments and evaluations on dance generation under different BPM music conditions, as well as comparisons with existing method for dance generation under in-the-wild music conditions. Additionally, we re-conducted subjective evaluation experiments based on the reviewers' suggestions. Given the discussion period has been extended by an additional six days, we are looking forward to engaging in further discussions with the reviewers to refine our work.

---

### Meta-Review · Area_Chair_tjFk · 2024-12-12

**Metareview:**

The paper receives mixed scores from the reviewers. While the reviewers appreciate the intuitive idea and well-organized descriptions, they also find several drawbacks including limited contributions and unconvincing analyses. In particular, quantitative evaluation (Table 2) shows that the proposed method performs worse using the DIV metric. The authors are encouraged to conduct more in-depth analyses to address these comments in the revised version.

**Additional Comments On Reviewer Discussion:**

During discussion, reviewer vNB2 asked for more evidence regarding generalization ability and performance analyses, while the authors showed some promising results, the proposed method performs inferior to existing methods on some quantitative metrics. Authors acknowledged that more analysis is required to understand the reason.

---

### Decision · Program_Chairs · 2025-01-22

Reject